

# New sources of *Sym2*$^A$ allele in the pea (*Pisum sativum* L.) carry the unique variant of candidate LysM-RLK gene *LykX*

Anton S. Sulima[1,*], Vladimir A. Zhukov[1,*], Olga A. Kulaeva[1], Ekaterina N. Vasileva[1], Alexey Y. Borisov[1] and Igor A. Tikhonovich[1,2]

[1] All-Russia Research Institute for Agricultural Microbiology, Saint-Petersburg, Russia
[2] Faculty of Biology, St. Petersburg State University, Saint-Petersburg, Russia
[*] These authors contributed equally to this work.

Corresponding author
Anton S. Sulima, ASulima@arriam.ru

## ABSTRACT

At the onset of legume-rhizobial symbiosis, the mutual recognition of partners occurs based on a complicated interaction between signal molecules and receptors. Bacterial signal molecules named Nod factors (''nodulation factors'') are perceived by the plant LysM-containing receptor-like kinases (LysM-RLKs) that recognize details of its structure (i.e., unique substitutions), thus providing the conditions particular to symbiosis. In the garden pea (*Pisum sativum* L.), the allelic state of *Sym2* gene has long been reported to regulate the symbiotic specificity: for infection to be successful, plants with the *Sym2*$^A$ allele (for ''*Sym2* Afghan'', as these genotypes originate mostly from Afghanistan) require an additional acetylation of the Nod factor which is irrelevant for genotypes with the *Sym2*$^E$ allele (for ''*Sym2* European''). Despite being described about 90 years ago, *Sym2* has not yet been cloned, though phenotypic analysis suggests it probably encodes a receptor for the Nod factor. Recently, we described a novel pea gene *LykX (PsLykX)* from the LysM-RLK gene family that demonstrates a perfect correlation between its allelic state and the symbiotic specificity of the *Sym2*$^A$-type. Here we report on a series of Middle-Eastern pea genotypes exhibiting the phenotype of narrow symbiotic specificity discovered in the VIR plant genetic resources gene bank (Saint-Petersburg, Russia). These genotypes are new sources of *Sym2*$^A$, as has been confirmed by an allelism test with *Sym2*$^A$ pea cv. Afghanistan. Within these genotypes, *LykX* is present either in the allelic state characteristic for cv. Afghanistan, or in another, minor allelic state found in two genotypes from Tajikistan and Turkmenistan. Plants carrying the second allele demonstrate the same block of rhizobial infection as cv. Afghanistan when inoculated with an incompatible strain. Intriguingly, this ''Tajik'' allele of *LykX* differs from the ''European'' one by a single nucleotide polymorphism leading to an R75P change in the receptor part of the putative protein. Thus, our new data are in agreement with the hypothesis concerning the identity of *LykX* and the elusive *Sym2* gene.

## INTRODUCTION

The mutualistic symbiosis formed by legume plants (family *Fabaceae*) and nodule bacteria (a diverse group of bacteria collectively called rhizobia) is an example of a highly specific interaction of symbiotic partners. Communication between plant and bacteria relies on an exchange of signal molecules, a process known as "molecular cross-talk" (*Oldroyd, 2001*; *Cooper, 2007*; *Zipfel & Oldroyd, 2017*). The roots of a legume plant excrete flavonoid compounds attracting rhizobia, that, in turn, begin to produce the bacterial signal molecules called Nod factors (Nodulation factors) (*Spaink, 1995*; *Schultze & Kondorosi, 1998*; *Perret, Staehelin & Broughton, 2000*; *Geurts & Bisseling, 2002*). Nod factors are lipochitooligosaccharide molecules that have specific decorations characteristic for a particular species, or even strains, of rhizobia. Legume plants possess receptors that are able to bind Nod factors in very low concentrations and recognize the fine structure of the ligands, including the specific moieties (*Bensmihen, De Billy & Gough, 2011*; *Delaux et al., 2015*; *Geurts, Xiao & Reinhold-Hurek, 2016*; also reviewed in *Gough et al., 2018*), so that only a perfectly fitting strain of rhizobia is allowed to enter the root through the root hairs. The successful recognition of a Nod factor activates a signal transduction cascade leading to a change in gene expression in root epidermal cells (*Larrainzar et al., 2015*; *Liu et al., 2019*), formation of bacterial microcolony on a tip of the root hair and growth of so-called "infection thread"; a structure inside the root hair through which rhizobia are delivered to root cells that form a nodule primordium (*Downie & Walker, 1999*; *Perret, Staehelin & Broughton, 2000*). If plant recognizes that the Nod factor of a particular bacterial strain does not fit, the aforementioned processes do not occur.

The phenomenon of the specificity of legume-rhizobial symbiosis was noticed at least 90 years ago, when the Russian geneticist *Govorov (1928)*, followed by Russian microbiologist *Razumovskaya (1937)*, described pea (*Pisum sativum* L.) specimens originating from Afghanistan as incapable of forming nodules in European soils (*Govorov, 1928*; *Razumovskaya, 1937*). Further on, some specimens of pea originating from several Middle East countries were described as failing to nodulate when interacting with rhizobia strains that are common in Europe (*Lie, 1971*; *Lie, 1978*; *Lie, Timmermans & Ladizinski, 1982*; *Lie & Timmermans, 1983*). These specimens, however, established symbiosis with strains isolated from the soils of Turkey, Israel and some other countries. Later it was found that all that strains were characterized by a specific modification of the Nod factor molecule, namely the additional acetylation at the reducing end of the chitin backbone in the C-6 position caused by the acetyltransferase nodX (*Davis, Evans & Johnston, 1988*; *Firmin et al., 1993*). Subsequently, in the work of *Ovtsyna et al. (1998)* it was shown that the gene *nodX* can be functionally replaced by the gene *nodZ* encoding a fucosyltransferase that fucosylates the reducing end of Nod factor.

This unusual plant nodulation trait was called the "Afghan" phenotype, since the typical genotype of pea with such a phenotypic manifestation originates from Afghanistan (today this genotype is known as cv. Afghanistan, JI1357 and NGB2150) (*Young & Matthews, 1982*; *Tsyganov et al., 1999*). In contrast, the trait of broad specificity—the ability to be nodulated by strains from both Europe and Middle East—was named the "European"

phenotype. From the plant side, the manifestation of the "Afghan" phenotype has been shown to be controlled by a single gene named *Sym2*, that was subsequently localized in the Ist linkage group of the pea genome (*Kozik et al., 1995*; *Kozik et al., 1996b*). Since rhizobia require the unique modification of the Nod factor molecule to successfully interact with the "Afghan" peas, it was assumed that *Sym2* may encode a Nod factor receptor with two different allelic states: $Sym2^A$ (from "Afghan", highly specific), and $Sym2^E$, or $Sym2^C$ (from "European" or "cultivated", less specific) (*Kozik et al., 1996a*; *Geurts et al., 1997*), although formal evidence of it is still lacking.

In 2003, genes encoding LysM-domain receptor-like kinases (LysM-RLKs) that probably perceive the Nod factor were described in the model legumes *Lotus japonicus* (Regel) Larsen (*NFR1* and *NFR5*) and *Medicago truncatula* Gaertn. (*LYK3*) (*Limpens et al., 2003*; *Madsen et al., 2003*; *Radutoiu et al., 2003*). Interestingly, *Limpens et al. (2003)* found a cluster of LysM-RLK genes in the *Sym2*-orthologous region of the *M. truncatula* genome and described one of them (*LYK3*) as the best candidate for encoding the Nod factor receptor. In 2008, Zhukov and colleagues discovered two pea homologs of *L. japonicus NFR1*, namely *Sym37* and *K1*, located in linkage group I, just where *Sym2* is located (*Zhukov et al., 2008*). Unfortunately, neither *Sym37* nor *K1* demonstrated any sequence features that could distinguish "Afghan" lines from "European" ones, although products of these genes still may play a role in determination of $Sym2^A$-type specificity as parts of the protein complexes. In our previous work (*Sulima et al., 2017*), a new pea gene named *LykX* (or *PsLykX*, for *Pisum sativum* LysM Kinase eXclusive) located in the same region of linkage group I was described. When analyzing the polymorphism of its first exon, a minor haplotype was found which is unique for lines with the "Afghan" phenotype. Thus, *LykX* has become a new candidate for the *Sym2*.

One of the drawbacks of many previous works aimed at finding the *Sym2* gene has been the use of a limited number of "Afghan" lines with high genetic uniformity; for example, they all possessed the same alleles of the *Sym37* and *K1* genes. This may be due to the fact that these lines are close relatives, or even descendants of the same original genetic line. More than 30 years ago, the study of wild varieties of pea from Afghanistan, Iran, Tibet and Turkey revealed seven varieties with the "Afghan" specificity, but no molecular-genetic analysis had been performed (*Kneen & LaRue, 1984*). In the present work, the similar approach was employed: we attempted to discover new "Afghan" forms among 122 pea samples originating from different regions of the Middle East. Our aim was to expand the sample of available "Afghan" lines, to evaluate the polymorphism of the *LykX* gene, and to see if it is associated with particular preferences towards rhizobia.

## MATERIALS & METHODS

### Plant material

The search for pea lines demonstrating the "Afghan" phenotype was conducted on the basis of material obtained from the VIR plant genetic resources gene bank (N. I. Vavilov All-Russian Institute of Plant Genetic Resources (VIR), St. Petersburg, Russia), kindly provided by Prof. M. A. Vishnyakova, head of Department of genetic resources of leguminous crops,

and E. V. Semenova, curator of the pea collection. The choice of material was determined by the geographical origin of samples (the Middle East). The whole sample contained 122 pea lines (see Table S1).

Additionally, a number of pea cultivars and genetic lines with the confirmed "Afghan" phenotype from the collection of the laboratory of genetics of plant-microbial interactions of the All-Russia Research Institute for Agricultural Microbiology (ARRIAM; St. Petersburg, Russia) were used as control (Table S2). The line 701 of *Pisum fulvum* Sibth. & Smith (Israel, Valley of the Cross) was kindly provided by Sc.D. O.E. Kosterin (Novosibirsk, Russia).

## Plant growth conditions

During the summer season of 2011, plants were grown in vegetation houses in conditions corresponding to the region's climate (St. Petersburg, Leningrad Oblast, Russia). When necessary for control, plants were grown in Vötsch Industrietechnik VB 1014 growth chambers (Balingen, Germany) under the following conditions:

- day/night: 16/8 hours;
- temperature: 21 °C;
- relative air humidity: 75%; illumination: 38000 lux.

The seeds were pre-sterilized with concentrated $H_2SO_4$ for between 10 and 15 minutes and then washed five times with distilled water. Sterilized seeds were germinated in Petri dishes filled with sterile vermiculite for 3 days at 27 °C, then seedlings were planted in pots. For large-scale experiments in vegetation houses, sterilization and preliminary germination of seeds was omitted.

Plants were grown in the 2 l pots filled with sand. When sterility was necessary, pots together with sand were heated in a dry-heat cabinet for 2 h at 200 °C. Mineral nutrition (Table S3) was added to substrate simultaneously with planting seedlings, 150 ml of solution per pot. When the analysis of the symbiotic phenotype was required, combined nitrogen in the form of ammonium nitrate was not added to the nutrient solution.

## Bacterial strains

In experiments involving analysis of nodulation, two contrast strains of *Rhizobium leguminosarum* bv. *viciae* were used:

Strain *R. leguminosarum* bv. *viciae* RCAM1026 (= CIAM1026), used in ARRIAM as a model strain (*Afonin et al., 2017*). This strain is unable to colonize a plant with the "Afghan" phenotype due to absence of the *nodX* gene ($nodX^-$ strain).

Strain *R. leguminosarum* bv. *viciae* A1, first isolated in the Leningrad Oblast from nodules of the "Afghan" pea line (*Chetkova & Tikhonovich, 1986*), thus possessing the *nodX* gene ($nodX^+$ strain).

For the visualization of symbiotic phenotype, the *gusA*-marked derivative of RCAM1026 (RCAM1026 *gusA*) with constitutive expression of *gusA* gene was used (*Kirienko et al., 2018*).

## Bacteria cultivation

The strains were cultivated on solid medium No. 79 (Table S4) in a thermostat at a temperature of 28 °C for three days. In the case of streptomycin-resistant strain RCAM1026, streptomycin (100 g/ml) was added to the medium; in the case of strain A1, no antibiotics were added. Plants were inoculated with a suspension of bacteria: the culture ("lawn") from one Petri dish was resuspended in 1 l of sterile distilled water (final CFU/ml : not less than $10^6$) and the resulting solution was added to the substrate while planting the seedlings, 100 ml per pot.

## Plant DNA extraction

DNA was extracted from young leaves (top or second-from-top node). Extraction was conducted according to the previously described CTAB protocol (*Rogers & Bendich, 1985*; *Sulima et al., 2017*).

## Sequencing of DNA fragments

To amplify the first exons of *LykX*, primers were used as described in *Sulima et al. (2017)*. The PCRs were performed in 0.5 ml eppendorf-type microcentrifuge tubes on an iCycler (Bio-Rad, Hercules, CA, USA) or Dyad (Bio-Rad) thermocycler using the ScreenMix-HS kit (Evrogen, Moscow, Russia). The PCR cycling conditions were as follows: 95 °C (5 min), 35 × [95 °C (30 s), Tm (varying depending on primers) (30 s), 72 °C (1 min)], 72 °C (5 min). The PCR fragments were sequenced using the ABI Prism3500xL system (Applied Biosystems, Palo Alto, CA, USA) at the "Genomic Technologies, Proteomics, and Cell Biology" Core Center of the ARRIAM (St.Petersburg, Russia). The resulting sequences have been deposited into the NCBI database under accession numbers MN187362–MN187364, MN200353–MN200358.

Analysis of sequences was conducted by means of MEGA7 software (*Kumar, Stecher & Tamura, 2016*). A Minimum Evolution tree was built using the modified Nei-Gojobori (assumed transition/transversion bias = 2) method (*Zhang, Rosenberg & Nei, 1998*). A bootstrap test was performed with 500 bootstrap replications. Alignments were done in MEGA7 and EMBL-EBI Clustal Omega web service (https://www.ebi.ac.uk/Tools/msa/clustalo/).

## Genotyping of F$_2$ population

To analyze the F$_2$ sample obtained from the cross between lines K-3821 and NGB2150 carrying different alleles of *LykX*, the CAPS (cleaved amplified polymorphic sequence) marker was designed using the dCAPS Finder 2.0 online service (*Neff, Turk & Kalishman, 2002*). The primers used were the same as for amplification of the 1$^{st}$ exon. The digestion of SNP (single nucleotide polymorphism) site was performed by the Hpy188III restriction enzyme (New England Biolabs, Massachusetts, USA).

## Microscopy

For detailed phenotype analysis, plants were grown in sand, as described above, while inoculated with suspension of RCAM1026 *gusA* culture. Four weeks after inoculation, 5 segments of roots 2–2.5 cm long were collected from each plant, leaving approximately two

cm from the base of the root and two cm from the tips (if possible). Root segments were stained for 18 h in a solution of 50 mM Na-phosphate buffer (pH 7.0) containing 0.1% Triton, 5 mM $K_3Fe(CN_6)$ (red blood salt), 5 mM $K_4Fe(CN_6)$ (yellow blood salt) and 0.02 % 5-bromo-4-chloro-3-indolyl-bD-glucuronic acid (X-glc). For storage, stained segments were fixated with 4% paraformaldehyde in a TBS buffer (50 mM TrisHCl, 150mM NaCl, pH 7.5) with the addition of 0.1% Tween-20 and 0.1% Triton X-100 under a vacuum (7 min, 3–4 times) using a VacuuBrand ME 1C vacuum pump (Vacuubrand, Wertheim, Germany) and then dehydrated in graded ethanol series (from 10% to 70%). For observation, fixated samples were treated with the same ethanol series in reverted order and then with TBS buffer. Microscopic analysis was conducted using Zeiss Axiovert 35 inverted phase contrast microscope (ZEISS, Oberkochen, Germany) by 100-fold magnification. The photos were taken on the Canon EOS 4000D camera (Canon, Tokyo, Japan).

## RESULTS

### Selection of pea lines with narrow symbiotic specificity

To increase the amount of the available "Afghan" lines, a sample of 122 pea lines originating from various regions of the Middle East (Table S1) was obtained from the VIR plant genetic resources gene bank. In the summer season of 2011, the nodulation phenotype of the lines was tested by inoculation with $nodX^+$ or $nodX^-$ strains. Five lines were discovered which did not form nodules when inoculated with $nodX^-$ strain RCAM1026, but demonstrated normal nodulation with strain A1 (i.e., the typical "Afghan" phenotype; so did plants of cv. Afghanistan (NGB2150) as well) (Table 1). From among the other lines analyzed, there were several with decreased or increased numbers of nodules formed by either the RCAM1026 ($nodX^-$) strain or A1 ($nodX^+$) strain, but these were not included in subsequent analysis.

### Allelism test between newly discovered and previously known $Sym2^A$-lines

In order to determine whether the symbiotic phenotype of selected pea accessions was caused by the same gene as in the previously known "Afghan" lines (i.e., the $Sym2$ gene), an allelism test was conducted. $F_1$ seeds obtained from the cross of previously reported "Afghan" lines NGB2150, K-6883 and K-6047-2 with lines selected from VIR gene bank, along with seeds of parental lines, were planted in pots with sterile sand and inoculated with the $nodX^-$ rhizobia strain RCAM1026 (see Materials and Methods). Additionally, two seeds of cv. Finale (characterized by the broad "European" symbiotic specificity and thus able to interact with the $nodX^-$ strain RCAM1026) were planted in each pot to confirm the success of the inoculation.

After 4 weeks of vegetation, $F_1$ hybrids, like the plants of parental lines, did not form nodules with the $nodX^-$ strain RCAM1026. Plants of cv. Finale in the same pots formed, on average, about 40 nodules per plant (see Table S5). The lack of nodules in $F_1$ plants indicates the allelism of genetic determinants of symbiotic phenotype in newly-described and previously known "Afghan" pea lines. Thus, new sources of $Sym2^A$ alleles have been identified: the lines K-1878, K-3374, K-3821, K-4902, and K-6559.
**Table 1  Specimens from VIR collection that did not form nodules with *nodX⁻* strain RCAM1026 ($n = 5$).**

| VIR accession number | Nodules formed with *R. leguminosarum* RCAM1026 (*nodX⁻*) | Nodules formed with *R. leguminosarum* A1 (*nodX⁺*), m ± SD | Place of origin |
|---|---|---|---|
| K-1878 | 0 | 58.0 ± 5.8 | Afghanistan |
| K-3374 | 0 | 61.6 ± 7.9 | Turkmenistan |
| K-3821 | 0 | 65.6 ± 4.9 | Tajikistan |
| K-4902 | 0 | 50.8 ± 3.5 | Uzbekistan |
| K-6559 | 0 | 75.4 ± 8.9 | Afghanistan |
| K-6566 | 0 | 94.2 ± 6.2 | Afghanistan |

### Sequencing of the first exon of *LykX* in the selected lines of pea

In our previous work (*Sulima et al., 2017*), a new LysM-RLK gene *LykX (PsLykX)* was identified as a promising candidate for the elusive gene *Sym2*, as it demonstrated the unique allelic state found only in two pea lines with the "Afghan" phenotype which leads to substitution of three amino acids in its corresponding protein: Q44R, N45Y, and A76D, all in the first LysM module of the receptor part. Here we sequenced the first exon of *LykX* (corresponding to the receptor domain of the putative protein) in the newly selected "Afghan" lines from the VIR gene bank, several lines with "European" phenotype from our laboratory collection, cv. Iran characterized by the leaky temperature-dependent "Afghan" phenotype (*Lie, 1971*; *Kozik et al., 1996a*), and a wild relative of garden pea, *Pisum fulvum*.

It was revealed that in the majority of lines with "Afghan" phenotype the $1^{st}$ exon of *LykX* has the same allelic state that was discovered previously (*Sulima et al., 2017*). In lines K-3821 and K-3374, however, *LykX* demonstrated another allelic state with unique substitutions observed neither in "European" nor in other "Afghan" lines. These lines were called "Tajik", according to the place of origin of line K-3821, which was identified first. Taken this new find into account, we suspected that the results of our initial allelism test might be ambiguous, since the similar non-nodulation phenotype in $F_1$ would be observed if the block of nodulation in the "Tajik" lines is caused by genetically dominant determinant independent of *Sym2*. In order to test that new hypothesis, the nodulation trait in $F_2$ progeny derived from the cross between "Tajik" line K-3821 and the typical "Afghan" line NGB2150 was analyzed in the inoculation experiment similar to the one described above. Of 119 $F_2$ plants, none demonstrated normal nodulation with RCAM1026 strain regardless of the allelic state of *LykX* (homozygous "Afghan" allele, homozygous "Tajik" allele, heterozygote), while control plants of cv. Finale formed, on average, 65.3 ± 20 (m ±SD, $n = 28$) nodules per plant (see Table S6 and Fig. S1). This suggests that our initial interpretation of allelism test results was correct.

The nucleotide substitutions unique for "Afghan" lines lead to the replacement of three amino acids in the protein in comparison to the "European" lines (Table 2). "Tajik" lines seem to be much more similar to the "European" ones, with only one unique amino acid substitution in the protein: R75P. It is important to note that all these characteristic substitutions are concentrated in the first LysM module (LysM1) of the receptor domain, which could indirectly indicate the importance of this particular module for the functioning of the LykX protein (Fig. 1).

**Table 2   Amino acid substitutions in the LykX protein characteristic for pea lines with different specificity of symbiosis.** "Afghan" variants are marked with green, "Tajik" with blue, "European" with red. *Pisum fulvum* line 701 is marked with yellow. New sources of *Sym2* alleles described in this work are presented in bold. Lines with narrow symbiotic specificity ("Afghan" phenotype) are framed.

| Pea line | Substitution position in protein | | | |
|---|---|---|---|---|
| | 44 | 45 | 75 | 76 |
| NGB2150 | R | Y | R | D |
| K-6883 | R | Y | R | D |
| K-6047-2 | R | Y | R | D |
| cv. Iran | R | Y | R | D |
| **K-1878** | R | Y | R | D |
| **K-4902** | R | Y | R | D |
| **K-6559** | R | Y | R | D |
| **K-3374** | Q | N | P | A |
| **K-3821** | Q | N | P | A |
| Caméor | Q | N | R | A |
| Finale | Q | N | R | A |
| SGE | Q | N | R | A |
| *P.fulvum* 701 | Q | N | R | A |

The plants of cv. Iran were found to carry the same allele of *LykX* as the cv. Afghanistan (NGB2150), which corresponds with the idea that their phenotype is determined by the same gene (*Kozik et al., 1995*; *Kozik et al., 1996a*). Interestingly, the "European" allele of *LykX* was found in *Pisum fulvum* line 701, the wild species closely related to *Pisum sativum*.

## Phenotype confirmation

According to previous descriptions (*Geurts et al., 1997*; *Zhukov et al., 2008*), "Afghan" peas are unable to form normally developed infection threads while inoculated with the $nodX^-$ strain of rhizobia. Bearing this in mind, we performed the microscopic analysis of roots of "Tajik" lines in order to characterize their symbiotic phenotype. Seedlings of the "Tajik" line K-3821 and the "Afghan" line NGB2150, along with the "European" cv. Caméor, were inoculated with *gusA*-marked strain CIAM1026. 4 weeks after inoculation, the root fragments were harvested, treated with staining buffer and observed under a light microscope. Both lines demonstrated normal root hair curling and formation of microcolonies, but infection was aborted at this stage so no infection threads were observed for neither "Afghan" nor "Tajik" lines on 25 cm of roots (Figs. 2B, 2C). Control plants of cv. Caméor developed normal infection threads (Fig. 2A). Accordingly, plants of cv. Caméor formed $81.6 \pm 15.2$ (m $\pm$SD, $n = 10$) nodules, while NGB2150 and K-3821 formed $0.2 \pm 0.6$ and $0.6 \pm 1.3$ nodules, respectively. Thus, we confirmed that "Afghan" and "Tajik" pea lines have an identical symbiotic phenotype, which is consistent with the alterations in the first exon sequences of *LykX* gene as compared to the "European" variant.

```
Cameor      MKLIFSLLLFLFLECVFFKVESKCVKGCDIALASYHVMPAFLLQNITNFMQSKIVSAFNS   60
Finale      MKLIFSLLLFLFLECFFSKVESKCVKGCDIALASYHVMPAFKLQNITNFMQSKIVSAFNS   60
SGE         MKLIFSLLFFLFVECVFFKVESKCVKGCDIALASYHVMPAFKLQNITNFMQSKIVSAFNS   60
Pf_701      MKLIFSLLLFLFVECFFSKVESKCVKGCDIALASYHVMPAFKLQNITNFMQSKIVSAFNS   60
K-3821      MKLIFSLLLFLFLECVFFKVESKCVKGCDIALASYHVMPAFKLQNITNFMQSKIVSAFNS   60
K-3374      MKLIFSLLLFLFLECVFFKVESKCVKGCDIALASYHVMPAFKLQNITNFMQSKIVSAFNS   60
NGB2150     MKLIFSLLLFLFLECVFFKVESKCVKGCDIALASYHVMPAFKLRYITNFMQSKIVSAFNS   60
K-6883      MKLIFSLLLFLFLECVFFKVESKCVKGCDIALASYHVMPAFKLRYITNFMQSKIVSAFNS   60
K-1878      MKLIFSLLLFLFLECVFFKVESKCVKGCDIALASYHVMPAFKLRYITNFMQSKIVSAFNS   60
K-6047-2    MKLIFSLLLFLFLECVFFKVESKCVKGCDIALASYHVMPAFKLRYITNFMQSKIVSAFNS   60
K-6559      MKLIFSLLLFLFLECVFFKVESKCVKGCDIALASYHVMPAFKLRYITNFMQSKIVSAFNS   60
K-4902      MKLIFSLLLFLFLECVFFKVESKCVKGCDIALASYHVMPAFKLRYITNFMQSKIVSAFNS   60
Iran        MKLIFSLLLFLFLECVFFKVESKCVKGCDIALASYHVMPAFKLRYITNFMQSKIVSAFNS   60
            ********:***:**.* ********************* *:. **************

Cameor      SDVLIRYNRDILSNRANIFSYFRVNVPFPCDCIGGEFLGHVFEYTANERDTYDLIANSYY  120
Finale      SDVLIRYNRDILSNRANIFSYSRVNIPFPCDCIGGEFLGHVFEYTANERDSYDLIANSYY  120
SGE         SDVLIRYNRDILSNRANIFSYFRVNIPFPCDCIGGEFLGHVFEYTANERDTYDLIANSYY  120
Pf_701      SDVLIRYNRDILSNRANIFSYFRVNIPFPCDCIGGEFLGHVFEYTANERDTYDLIANSYY  120
K-3821      SDVLIRYNRDILSNPANIFSYFRVNIPFPCDCIGGEFLGHVFEYTANERDTYDLIANSYY  120
K-3374      SDVLIRYNRDILSNPANIFSYFRVNIPFPCDCIGGEFLGHVFEYTANERDTYDLIANSYY  120
NGB2150     SDVLIRYNRDILSNRDNIFSYFRVNIPFPCDCIGGEFLGHVFEYTANERDTYDLIANSYY  120
K-6883      SDVLIRYNRDILSNRDNIFSYFRVNIPFPCDCIGGEFLGHVFEYTANERDTYDLIANSYY  120
K-1878      SDVLIRYNRDILSNRDNIFSYFRVNIPFPCDCIGGEFLGHVFEYTANERDTYDLIANSYY  120
K-6047-2    SDVLIRYNRDILSNRDNIFSYFRVNIPFPCDCIGGEFLGHVFEYTANERDTYDLIANSYY  120
K-6559      SDVLIRYNRDILSNRDNIFSYFRVNIPFPCDCIGGEFLGHVFEYTANERDTYDLIANSYY  120
K-4902      SDVLIRYNRDILSNRDNIFSYFRVNIPFPCDCIGGEFLGHVFEYTANERDTYDLIANSYY  120
Iran        SDVLIRYNRDILSNRDNIFSYFRVNIPFPCDCIGGEFLGHVFEYTANERDTYDLIANSYY  120
            **************  *****  ***:******************************:*********

Cameor      ASLTSVQVLQKFNSYHPNHIPAKAKVNVTVNCSCGNSQISKDYGLFITYPLRSTDSLEKI  180
Finale      ASLTSVQVLQKFNSYHPNHIPIKAKVNVTVNCSCGNSQISKDYGLFITYPLRSTDSLEKI  180
SGE         ASLTSVQVLQKFNSYHPNHIPIKAKVNVTVNCSCGNSQISKDYGLFITYPLRSTDSLEKI  180
Pf_701      ASLTSVQVLQKFNSYHPNHIPAKAKVNVTVNCSCGNSQISKDYGLFITYPLRSTDSLEKI  180
K-3821      ASLTSVQVLQKFNSYHPNHIPIKAKVNVTVNCSCGNSQISKDYGLFITYPLRSTDSLEKI  180
K-3374      ASLTSVQVLQKFNSYHPNHIPIKAKVNVTVNCSCGNSQISKDYGLFITYPLRSTDSLEKI  180
NGB2150     ASLTSVQVLQKFNSYHPNHIPIKAKVNVTVNCSCGNSQISKDYGLFITYPLRSTDSLEKI  180
K-6883      ASLTSVQVLQKFNSYHPNHIPIKAKVNVTVNCSCGNSQISKDYGLFITYPLRSTDSLEKI  180
K-1878      ASLTSVQVLQKFNSYHPNHIPIKAKVNVTVNCSCGNSQISKDYGLFITYPLRSTDSLEKI  180
K-6047-2    ASLTSVQVLQKFNSYHPNHIPIKAKVNVTVNCSCGNSQISKDYGLFITYPLRSTDSLEKI  180
K-6559      ASLTSVQVLQKFNSYHPNHIPIKAKVNVTVNCSCGNSQISKDYGLFITYPLRSTDSLEKI  180
K-4902      ASLTSVQVLQKFNSYHPNHIPIKAKVNVTVNCSCGNSQISKDYGLFITYPLRSTDSLEKI  180
Iran        ASLTSVQVLQKFNSYHPNHIPIKAKVNVTVNCSCGNSQISKDYGLFITYPLRSTDSLEKI  180
            ********************  ********************************************

Cameor      ANAFKLDEGLIQNFNPDVNFSKGSGIVFIPGR      212
Finale      ANASKLDEGLIQNFNPDVNFSKGSGIVFIPGR      212
SGE         ANASKLDEGLIQNFNPDVNFSKGSGIVFIPGR      212
Pf_701      ANASKLDEGLIQNFNPDVNFSKGSGIVFIPGR      212
K-3821      ANASKLDEGLIQNFNPDVNFSKGSGIVFIPGR      212
K-3374      ANASKLDEGLIQNFNPDVNFSKGSGIVFIPGR      212
NGB2150     ANASKLDEGLIQNFNPDVNFSKGSGIVFIPGR      212
K-6883      ANASKLDEGLIQNFNPDVNFSKGSGIVFIPGR      212
K-1878      ANASKLDEGLIQNFNPDVNFSKGSGIVFIPGR      212
K-6047-2    ANASKLDEGLIQNFNPDVNFSKGSGIVFIPGR      212
K-6559      ANASKLDEGLIQNFNPDVNFSKGSGIVFIPGR      212
K-4902      ANASKLDEGLIQNFNPDVNFSKGSGIVFIPGR      212
Iran        ANASKLDEGLIQNFNPDVNFSKGSGIVFIPGR      212
            ***  ***************************
```

**Figure 1   Alignment of the putative receptor domain sequences of LykX protein from lines represented in Table 2.** LysM modules are highlighted by color (orange, LysM1; green, LysM2; violet, LysM3). Sequences corresponding to the lines with the *Sym2*$^A$-type symbiotic specificity are boxed. Sites that distinguish "Afghan", "European" and "Tajik" variants of LykX are given in bold.

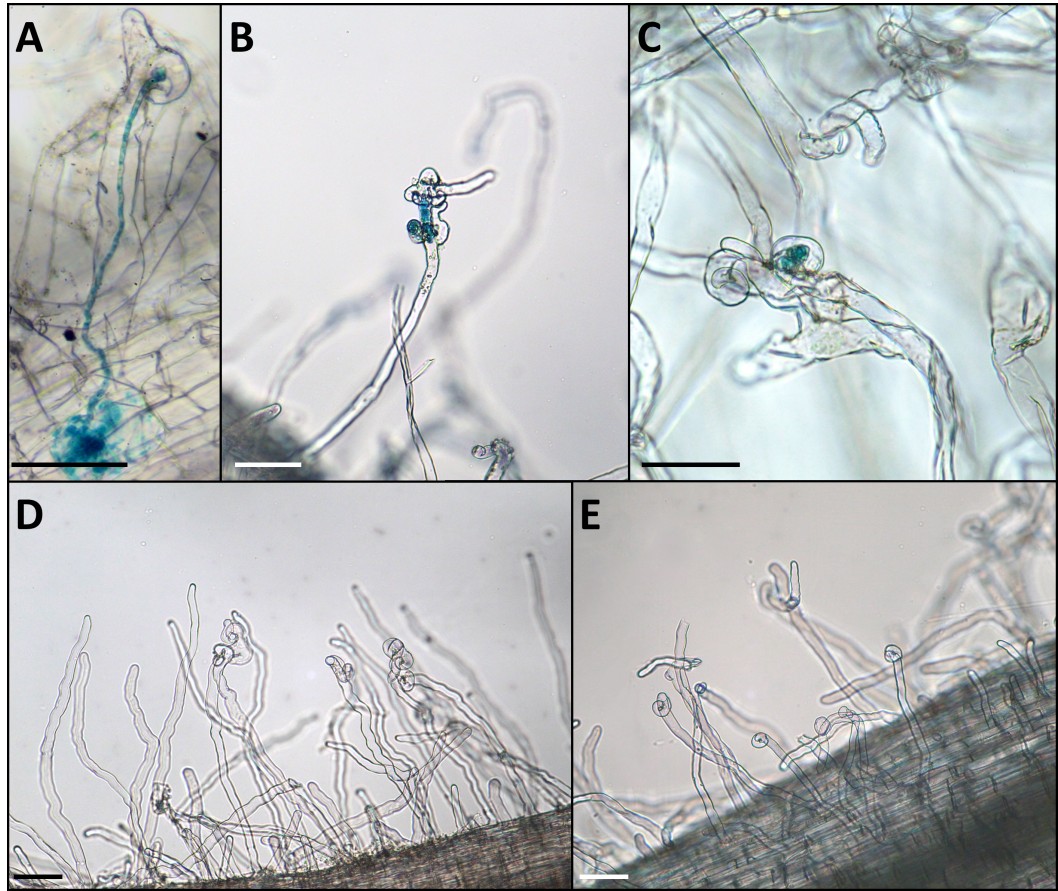

**Figure 2 Phenotype of the infection threads in plants with different *LykX* alleles in the presence of *nodX⁻* strain RCAM1026 *gusA*.** (A) normal infection thread formed by the "European" line Caméor. (B, C) early abortion of the infection thread in "Afghan" line NGB2150 (B) and "Tajik" line K-3821 (C). (D, E) multiple curled root hairs of "Aghan" (D) and "Tajik" (E) plants. In most cases, the infection does not go beyond this stage. Scale bares are 0.1 mm.

# DISCUSSION

Legume-rhizobial symbiosis relies on proper mutual recognition of partners, a process involving signal molecules and receptors that have been evolving step-by-step in a fashion similar to an "arms race". Indeed, a plant needs to limit the set of bacteria penetrating its tissues as part of its defenses from pathogens and/or incompatible symbionts. On the other hand, bacteria may benefit when they broaden the range of possible hosts. Consequently, plants have evolved highly specific receptors that allow for discrimination of compatible and incompatible bacteria; in turn, rhizobia develop more and more variants of signal molecules (*Price et al., 1992*) that may be recognized by highly discriminatory plant receptors.

The pea is a unique object for studying the specificity of plant-microbe interactions due to the existence of a distinct group of specimens from Afghanistan showing a particular receptivity towards the microsymbiont. Such a phenotype is characteristic for plants

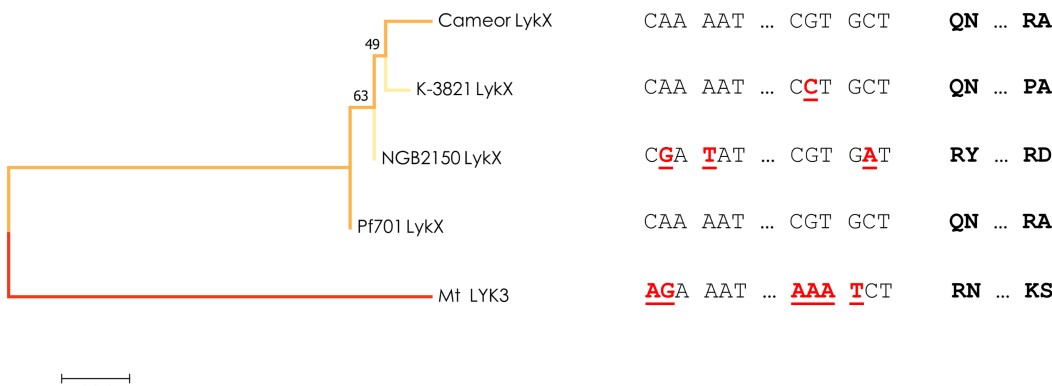

**Figure 3** **Estimated evolutionary relationships between members of IRL clade of legumes based on sequence of the part of *LykX* gene or its orthologue corresponding to LysM1 module.** The evolutionary history was inferred using the Minimum Evolution method. The optimal tree with the sum of branch length = 0.640 is shown. On the right are the SNPs that distinguish the alleles and the amino acids corresponding to them. Pf701 : the LysM1 module of *LykX* gene in *Pisum fulvum* line 701. Mt LYK3 : the LysM1 module of *LYK3* gene in *Medicago truncatula* Gaertn., the close orthologue of *LykX*.

carrying the *Sym2^A* allele conferring "resistance" to nodulation with rhizobia strains producing the Nod factor without the required decoration. According to genotyping data (*Jing et al., 2010*), Afghanistan peas represent a distinct clade remote from both modern garden pea varieties and *Pisum fulvum*, the wild relative to *Pisum sativum*. Consistent with this, the recently-discovered LysM-RLK gene *LykX* is present among peas from Afghanistan and adjacent countries in a unique form that encodes a protein differing by three amino acid residues from that of "European" peas and of *Pisum fulvum* (Table 2). This unique haplotype is associated with a specific "Afghan" nodulation phenotype, which makes *LykX* a good candidate for the *Sym2* gene. The "Afghan" haplotype of *LykX* was found in 3 previously uncharacterized lines originating from Afghanistan and Uzbekistan.

Interestingly, a different allele of *LykX* was discovered in two lines from Tajikistan and Turkmenistan manifesting the same strict symbiotic phenotype. To date we can't confirm that a single nucleotide swap resulting in R75P amino acid substitution in the "Tajik" variant of LykX protein actually affects its functionality in a way that leads to a narrow receptivity, but proline is known by its increased conformational rigidity which indeed may have a significant effect on the secondary structure of the protein (*Biedermannova et al., 2008*). Yet, in the light of our recent findings, we can speculate that especially "choosy" forms of symbiosis may sometimes be an adaptive trait, as the "Tajik" allele of *LykX* arose from the "European" variant independently and more recently than the "Afghan" one, which is supported by a minimum-evolution tree constructed using the modified Nei-Gojobori method (Fig. 3). However, the reasons why peas growing in this particular region (the Middle East) had to narrow down their symbiotic specificity several times remain unclear. It can be assumed that certain environmental conditions (including the behavior of the native soil microbiota) forced them to more strictly limit the range of bacteria that have access to their root tissue.

*Pisum fulvum*, which we used as an outgroup in the study, possesses the "European" variant of *LykX*, implying that it might form nodules with *nodX⁻* strains of *Rhizobium leguminosarum*. However, *P. fulvum* has been previously reported to form ineffective nodules with European strains (*Lie, 1981*), so it is likely that the infection stage of *P. fulvum* with *nodX⁻* strains may proceed normally. Another noteworthy pea accession in this work is cv. Iran, which, according to previous data, possesses the *Sym2$^A$* allele (*Kozik et al., 1995*) and, as shown in current work, carries the "Afghan" allele of *LykX*. In our experiments it did not nodulate with the *nodX⁻* strain, although previously it has been documented that cv. Iran can form nodules with *nodX⁻* strains under high temperatures (28 °C) (*Lie, 1971*; *Kozik et al., 1996a*). Using *Sym2$^A$* introgression lines, Kozik and colleagues have shown that temperature-dependent phenotype is intrinsic to cv. Afghanistan (NGB2150) as well, but is somewhat masked by the native genetic background (*Kozik et al., 1996a*). Our findings so far are in concert with these data, but the phenomenon in general may require further investigation. Moreover, there might be a chance of finding lines similar to cv. Iran in our sample of 122 pea accessions, as there are some lines with decreased numbers of nodules, and the temperatures in the Leningrad Oblast reached up to 30 °C during the summer of 2011.

Thus, our data are in agreement with the hypothesis of the identity of *Sym2* and *LykX*. Accordingly, it seems appropriate to name the newly discovered allele of *LykX Sym2$^T$* (for "Tajik"). However, genetic transformation experiments are still necessary to conclusively confirm or reject this hypothesis. It would also be essential to study the interaction of "Tajik" lines with rhizobial strains producing fucosylated Nod factor due to the presence of the *nodZ* gene, as their range of specificity may differ from that of the "typical Afghan" lines.

## CONCLUSIONS

In this work we identified a series of naturally-occurred forms of pea (*Pisum sativum* L.) with the narrow symbiotic specificity similar to that of cv. Afghanistan, and confirmed that their phenotype is prompted by the same genetic determinant *Sym2*. It is highly likely that the recently discovered gene *LykX* is in fact the elusive *Sym2* gene. Indeed, we found that within our sample, *LykX* demonstrates two allelic states ("Tajik" and "Afghan") different from those found in forms with normal symbiotic receptivity ("European"). Both allelic states lead to different amino acid substitutions in their corresponding proteins, as compared to the more common "European" state. In neither *K1* nor *Sym37* –the previous candidate genes for *Sym2* – were observed such correlations between allelic states and the "Afghan" phenotype. The results of our work also highlight the importance of the strict specificity of symbiosis for the pea, since, according to our data, it appeared independently at least twice during the evolution of *Pisum sativum* species.

## ACKNOWLEDGEMENTS

Part of our research (Sanger sequencing, microscopy) was performed using equipment of the Core Center "Genomic Technologies, Proteomics and Cell Biology" at the All-Russia

Research Institute for Agricultural Microbiology (ARRIAM), St. Petersburg, Russia. Biological material (pea seeds) were kindly provided by Prof. M. A. Vishnyakova (N. I. Vavilov All-Russian Institute of Plant Genetic Resources (VIR), St. Petersburg, Russia), E. V. Semenova (N. I. Vavilov All-Russian Institute of Plant Genetic Resources (VIR), St. Petersburg, Russia) and Sc.D. O.E. Kosterin (Institute Of Cytology And Genetics, Novosibirsk, Russia). The RCAM1026 *gusA* strain of *Rhizobium leguminosarum* bv. *viciae* was kindly provided by Dr. G.A. Akhtemova (All-Russia Research Institute for Agricultural Microbiology (ARRIAM), St.Petersburg, Russia).

### Funding

This work was financially supported by the Russian Science Foundation (project No. 17-76-30016). The funders had no role in study design, data collection and analysis, decision to publish, or preparation of the manuscript.

### Grant Disclosures

The following grant information was disclosed by the authors:
Russian Science Foundation: 17-76-30016.

### Competing Interests

The authors declare there are no competing interests.

### Author Contributions

- Anton S. Sulima conceived and designed the experiments, performed the experiments, analyzed the data, prepared figures and/or tables, authored or reviewed drafts of the paper, approved the final draft.
- Vladimir A. Zhukov conceived and designed the experiments, analyzed the data, contributed reagents/materials/analysis tools, prepared figures and/or tables, authored or reviewed drafts of the paper, approved the final draft.
- Olga A. Kulaeva performed the experiments, contributed reagents/materials/analysis tools, authored or reviewed drafts of the paper, approved the final draft.
- Ekaterina N. Vasileva performed the experiments, prepared figures and/or tables, authored or reviewed drafts of the paper, approved the final draft.
- Alexey Y. Borisov conceived and designed the experiments, contributed reagents/-materials/analysis tools, authored or reviewed drafts of the paper, approved the final draft.
- Igor A. Tikhonovich conceived and designed the experiments, authored or reviewed drafts of the paper, approved the final draft.

### DNA Deposition

The following information was supplied regarding the deposition of DNA sequences:
The LysM-RLK gene LykX first exon sequences are available at the NCBI database: MN187362–MN187364, MN200353–MN200358.

## Data Availability

The whole list of pea accessions obtained from VIR plant genetic resources gene bank is available in a Supplemental File.

## Supplemental Information

Supplemental information for this article can be found online at http://dx.doi.org/10.7717/peerj.8070#supplemental-information.

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
