# Peer review of "New sources of Sym2A allele in the pea (Pisum sativum L.) carry the unique variant of candidate LysM-RLK gene LykX"

_PeerJ, doi:10.7717/peerj.8070_

## Round 0.1 · original submission · Major Revisions

The reviewers have some critical comments demanding revision. However it is close to minor revision. Please check validity of the findings and modern literature citation, as suggested by first reviewer. It is important to include a segregation analysis of the new "Tajik" allele. Additional polishing of English presentation is welcome.

·

Basic reporting

The study presented here aimed (i) to identify additional pea (Pisum sativum) accessions that harbour the Sym2A allele, and (ii) to determine whether non-synonymous polymorphisms in the LysM-receptor encoding gene PsLykX correlate with this allele.
Pea Sym2 is a locus that determines rhizobium Nod factor structure-dependent host specificity. Pea accessions that harbour the so-called Afghan allele of Sym2 (Sym2A) can only be effectively nodulated by Rhizobium leguminosarum biovar viciae (Rlv) strains that contain nodX, a gene encoding an acetyl-transferase that acetylates that reducing end of the Nod factors.

The molecular nature of the pea Sym2 locus is an enigma in the legume symbiosis research field. It is generally anticipated that it encodes a LysM-type receptor kinase that functions as a Nod factor receptor, as such genes have been identified in a Sym2 syntenic region in Medicago truncatula (Limpens et al., 2003). However, formal evidence that this is the case is lacking. I have the opinion that this should be stated more clearly in the manuscript.

Also, it remains elusive whether the Sym2 locus represents single or multiple genes. For example, in M. truncatula and Lotus japonicus at least two tandemly duplicated LysM-type RLK encoding genes have been identified in the Sym2 orthologous region that function in LCO recognition (Limpens et al., 2003; Murakami et al., 2018). In pea at least 3 LysM-RLKs (PsK1, PsSym37 and PsLykX) have been found to be genetically linked to the Sym2 locus. So, it can not be ruled out that the LCO-dependent infection phenotype is determined by the interplay of two or more genes.

The reason why the molecular nature of the Sym2 locus has not yet been resolved is two-fold; (i) the complexity of the pea genome which frustrates genome sequencing strategies, and (ii) the difficulty to conduct reverse genetic studies due to extremely inefficient transformation protocols. Sulima et al., identified additional pea lines that harbour a Sym2A-like allele, are of relevance for future genome sequencing initiatives.

I find the manuscript well structured and written in clear and understandable English. Although I am not a native speaker, I have the feeling that editing by a native English speaker may be welcomed to further improve the MS.

Concerning literature references, most of the relevant papers have been cited. However, in my opinion, one citation, which is critical to understand the Nod factor specificity of Sym2A is lacking; namely, Ovtsyna et al., 1998 Mol. Plant-Microbe Interact. In this study, it is shown that the bacterial nodX gene (encoding an acetyl-transferase) can be functionally replaced by nodZ of Bradyrhizobium to allow nodulation on SYM2A harbouring pea lines. As nodZ encodes a fucosyl-transferase that fucosylates the reducing terminus of Nod factors, it clearly indicates that the Nod factor specificity encoded by Sym2A is less stringent than suggested here by Sulima et al. (e.g. lines 91-94).

Experimental design

Sulima and co-workers identified 6 pea accessions that can be nodulated with the nodX harbouring Rlv strain A1, but not with the nodX lacking strain RCAM1026. Subsequent allelism studies were conducted using a complex crossing scheme (Table S5) and nodulation assays on F1 hybrid plants. Although I can agree with the chosen strategy of crossings, I feel uncomfortable with the low number of F1 plants that have been tested (as low as n=3). Also, it can not be ruled that the inhibitory infection phenotype in the newly identified accessions is genetically dominant, though independent of Sym2. Therefore, it will be needed to study also an F2 population.

Validity of the findings

Based on the finding of non-synonymous polymorphisms in the receptor domain of PsLykX it is argued that “it is very likely PsLykX is Sym2” (line 307-311). To my opinion, the scientific basis for such a hypothesis is lacking. The authors identified a new allele for PsLykX (named Tajik), which is present in two of the four newly identified Sym2A harbouring accessions (K-3821 and K-3374). However, there is no amino acid substitution identified that strictly correlates with the Sym2A allele. To overcome this problem, it is argued that the Afghan and Tajik allelic variants of PsLykX evolved in parallel, resulting in different amino acid substitutions when compared to the ‘European’ alleles of PsLykX (line 311-313). To me, this parallel evolution hypothesis seems far fetched and would require additional support. Possibly the authors could make use of trans complementation studies of Lotus japonicus receptor mutants and the observation that Sym2A can b nodulation also by a nodZ harbouring strain (which is generally present in Mesorhizobium loti, the natural host of L. japonicus).

Reviewer 2 ·

Basic reporting

In this manuscript Sulima et al determined the nodulation phenotype of 122 pea lines from the Middle East inoculated with nodX- and nodX+ R. leguminosarum strains in order to find new “Afghan” lines requiring acetylation in the C6 position of the Nod factor. In total six new lines were found. Subsequent sequencing of intron 1 of the LykX gene in these lines revealed SNPs leading to three amino acid substitutions previously found in this gene within “Afghan” lines. Interestingly two pea lines had a single nucleotide change that leads to a single amino acid substation compared to the “European” lines. This new “Tajik” allele was by allelism test shown to be allelic to the previous Sym2A allele in “Afghan” lines and also result in the same lack of infection tread development.

The manuscript is generally well-written and the results presented supports the conclusions.

In line 63 a review from Gough et al 2018 is used to summarize crucial observations. This is not very helpful for the reader and at least some of the original articles should be cited.

Line 456. The journal listed for this work is wrong.

Experimental design

The experimental setup presented is sound and the methodology well described. However, the genetics of the new “Tajik” allele is not sufficiently described. It is important to determine whether this allele is a recessive, dominant or semidominant. One would assume that the “Tajik” line has already been crossed with an appropriate line and nodulation of a segregating population has been or could be scored.

Table 1 list a K-6566 line but no sequence analysis of the LykX gene is included in Table 2. What is the reason for this omission?

Validity of the findings

The genetics of the new “Tajik” allele is not sufficiently described. It is important to determine whether this allele is a recessive, dominant or semidominant. One would assume that the “Tajik” line has already been crossed with an appropriate line and nodulation of a segregating population has been or could be scored.

---

## Round 0.2 · accepted · Accept

The reviewer had no more critical comments demanding revision. Just note “Tajik” X “European” hybrid in discussion, if possible. I believe the manuscript have to be accepted now.

Reviewer 2 ·

Basic reporting

ok

Experimental design

could be improved

Validity of the findings

ok

Additional comments

I still think that analysis of a segregating population from a “Tajik” X “European” hybrid, that the authors say they will do in the future, would have allowed them to make a stronger conclusion on the current manuscript.